# From Biology to Diagnosis and Treatment: The Ariadne’s Thread in Cancer of Unknown Primary

**DOI:** 10.3390/ijms24065588

**Published:** 2023-03-15

**Authors:** Beatrice Gadiel Mathew, Fine Aliyuda, Denis Taiwo, Kehinde Adekeye, Godwin Agada, Elisabet Sanchez, Aruni Ghose, Elie Rassy, Stergios Boussios

**Affiliations:** 1Department of Medical Oncology, Medway NHS Foundation Trust, Windmill Road, Gillingham ME7 5NY, UK; 2Department of Medical Oncology, Mount Vernon Cancer Centre, East and North Hertfordshire NHS Trust, London HA6 2RN, UK; 3Department of Medical Oncology, Barts Cancer Centre, St. Bartholomew’s Hospital, Barts Health NHS Trust, London EC1A 7BE, UK; 4Department of Medical Oncology, Institut Gustave Roussy, 94805 Villejuif, France; 5King’s College London, Faculty of Life Sciences & Medicine, School of Cancer & Pharmaceutical Sciences, London SE1 9RT, UK; 6Kent Medway Medical School, University of Kent, Canterbury CT2 7LX, UK; 7AELIA Organization, 9th Km Thessaloniki—Thermi, 57001 Thessaloniki, Greece

**Keywords:** cancer of unknown primary (CUP), biology, molecular profiling, classification, diagnosis, treatment

## Abstract

Cancer of unknown primary (CUP) encloses a group of heterogeneous tumours, the primary sites for which cannot be identified at the time of diagnosis, despite extensive investigations. CUP has always posed major challenges both in its diagnosis and management, leading to the hypothesis that it is rather a distinct entity with specific genetic and phenotypic aberrations, considering the regression or dormancy of the primary tumour; the development of early, uncommon systemic metastases; and the resistance to therapy. Patients with CUP account for 1–3% of all human malignancies and can be categorised into two prognostic subsets according to their clinicopathologic characteristics at presentation. The diagnosis of CUP mainly depends on the standard evaluation comprising a thorough medical history; complete physical examination; histopathologic morphology and algorithmic immunohistochemistry assessment; and CT scan of the chest, abdomen, and pelvis. However, physicians and patients do not fare well with these criteria and often perform additional time-consuming evaluations to identify the primary tumour site to guide treatment decisions. The development of molecularly guided diagnostic strategies has emerged to complement traditional procedures but has been disappointing thus far. In this review, we present the latest data on CUP regarding the biology, molecular profiling, classification, diagnostic workup, and treatment.

## 1. Introduction

Cancer of unknown primary origin (CUP) is a diverse category of cancers with varying clinical and histological characteristics for which no original tumour site has been identified despite a thorough diagnostic workup [1]. It is the seventh to the eighth most frequent malignant disease for both sexes, whilst it has risen to become the fourth most common cause of cancer-related death, accounting for 1–3% of all human cancers [2]. 

Recently, a consensus was reached on the first diagnostic layer for CUP. History and physical examination; full blood count; along with serum markers, a CT scan of the chest/abdomen/pelvis, and a biopsy of the most accessible lesion, followed by immunohistochemical testing should be the starting point. On the other hand, the symptom-guided magnetic resonance imaging (MRI) or ultrasound, a positron emission tomography/computerised tomography (PET/CT) scan, targeted gene panels, immunohistochemical markers, and whole genome sequencing remain debatable [3]. With regard to organisational recommendations, namely, the National Comprehensive Cancer Network (NCCN), American Society of Clinical Oncology (ASCO), European Society for Medical Oncology (ESMO), and Spanish Society of Medical Oncology (SEOM), the histological evaluation is standard and is based on morphology and algorithmic immunohistochemistry (IHC) [4]. Traditionally, patients with CUP can be categorised into two prognostic groups according to their clinical and pathologic presentations. Those with a constellation of manifestations that can be assigned to a primary account for around 15–20% of CUP and are treated accordingly. The remaining patients have an unfavourable prognosis and are commonly treated with empirical chemotherapy [1,5,6,7]. The research in the field of CUP is mainly directed towards the development of molecular diagnostics to facilitate accurate prediction of the primary site, rather than to further investigate the already existing chemotherapeutics. This review aims to highlight different aspects around CUP such as the biology, clinicopathological subsets, diagnostic work-up, and therapeutic strategies.

## 2. Epidemiology of CUP

From the epidemiological perspective, around 8600 new CUPs are diagnosed each year in the United Kingdom, making this the fifteenth most prevalent cancer [8]. In 2012, Denmark had around 338 new cases per 100,000 people, compared to 284 in Germany, 296 in Canada, and 318 in the United States [9]. Since the early 1980s, CUP incidence rates in the United States have been falling at a pace of 3.6% per year over the previous two decades. Since 1973, the rate of non-microscopically confirmed CUP has decreased by 2.6% every year [10]. CUP incidence rates in Scotland climbed from 7 to 8 per 100,000 in the 1960s to a peak of 14–18 per 100,000 in the early to mid-1990s, before falling precipitously to 8 per 100,000 in 2009 [11]. Between 1999 and 2017, the incidence of CUP decreased in Korea. CUP incidence probably decreased due to improved diagnostics, which have led to better identification of the primary culprit. Genomic profiling testing may help in identifying molecular signatures in CUP patients and enable targeted treatment [12]. Patients aged 80 and up had the highest risk of occurrence. The survival rate climbed from 14.2% in 1999–2002 to 27.3% in 2013–2017 [13]. In Sweden, it is estimated that the age-standardised incidence increased from 10 per 100,000 in the early 1960s to 16 per 100,000 around the year 2000 for both males and females [14].

## 3. Risk Factors of CUP

Any degree of smoking raises the chance of CUP with respiratory system metastases to 4.9%. In a study, smoking was strongly linked to an increased risk of CUP, with a relative risk of 3.66% for ongoing, heavy smokers (>25 cigarettes/day) compared to never smokers (standardised for other risk factors), and a relative risk of 5.12% for patients with CUP who died within less 12 months since the diagnosis [15]. Other risk factors include alcohol intake, body mass index (BMI), waist circumference, diabetes, and a poor educational level or socioeconomic position [16]. The pathophysiology of familial CUP is characterised by the existence of a genetic vulnerability that puts family members of a CUP patient at a higher risk of CUP and other tumours [17]. Relatives of CUP patients are more likely to develop CUP as well as other malignant cancers, especially of the lung, pancreas, and colon [18].

## 4. Biology of CUP

Generally, the ongoing improved knowledge of the biology of several cancers has enabled a more-accurate classification, diagnosis, and prognosis, as well as providing guidance in the tailoring of specific therapies. However, in CUP, the data are not very mature. The early dissemination of tumour cells implies subsequent independent progression of the primary tumour and metastases, under the selection pressure of the immune system. The clinical observation of the high systemic relapse rate in CUP patients with localised disease treated with curative intention with surgery and/or radiotherapy supports that model. Still, molecular platforms represent a key component of the diagnostic work-up and clinical management [12]. 

### 4.1. Chromosomal Abnormalities

Many tumours have an abnormal number of chromosomes, a condition known as aneuploidy. Chromosomal instability (CIN), a mechanism of continual chromosomal missegregation, is a common cause of aneuploidy [19]. Oncogenesis necessitates huge numbers of genetic changes that cannot be caused by the regular rate of mutation, necessitating some sort of innate genomic instability to produce a mutator phenotype [20,21]. In a survey of 152 individuals with metastatic adenocarcinoma or undifferentiated CUP, 106 (70%) had clusters of cells with aberrant cellular DNA composition, while the rest were diploid. The prevalence of aneuploidy was comparable between sexes and had no discernible link with the various forms of metastatic invasion [22].

CIN is a known driver of early dissemination and aggressive behaviour of CUP. In a recently published study, researchers investigated the genomic information of CUP samples analysed using a hybridisation-capture-based next-generation sequencing (NGS) assay in 410 cancer-associated genes [23]. CUP samples presented mainly a very low aneuploidy score (AS) (63.0%; *n* = 92), followed by intermediate and low AS (16.4%; *n* = 24 each) and high AS (4.1%; *n* = 6). The high-intermediate AS groups lacked an enriched genetic alteration, and the presence of a *TP53* or *KRAS* mutation did not correlate with a high AS. This differs from what was previously known about these mutations in aneuploid cancers. The researchers concluded that CUP patients present individual gene alterations implicated in immune evasion and resistance to ICI, but further clinical investigations are needed to clarify the interplay between CIN, point mutations, and the immune system. 

### 4.2. Oncogenes and Proteins

Oncogenes play a critical role in cancer formation by either overexpression or amplification. The occurrence of protein overexpression or oncogene gene alterations in CUP is comparable to rates observed in metastatic cancers of known primary origin, with the expected variability [21]. *PI3K*, *Ras*, *p53*, *PTEN*, *Rb*, and *p16INK4a* are among the oncogenes and tumour suppressors that are often mutated in cancerous cells [24]. The mutation trends of the tumour suppressor *TP53* (which encodes p53), ataxia telangiectasia mutated (*ATM*), and cyclin-dependent kinase inhibitor 2A (*p16INK4A* and *p14ARF* encoded by CDKN2A) corroborate the oncogene-induced DNA replication stress concept [25]. Studies that assessed archival tumour tissues with NGS revealed alterations of *TP53* (38–55%), *Ras* (18–20%), *CDKN2A* (19%), *MYC* (12%), *ARID1A* (11%), and *PIK3CA* (9–14%) [26]. EGFR is widely expressed in CUP (74–75%), according to immunohistochemical studies, but c-KIT and HER2/neu are seldom active (overexpression in 4–27%) [20]. No meaningful link has been found between EGFR expression level and patient outcomes [27]. However, the expression of EGFR is associated with sensitivity to platinum-based regimens. CUP patients with overexpression of HER2/neu have mostly supradiaphragmatic disease, whereas histologically they are predominantly poorly differentiated adenocarcinomas. Given that *HER2/neu* amplifications have not been identified as driver mutations, very little response data have been published in HER2/neu-altered CUP.

Regardless of the fact that RAS-pathway-activating mutations are described in approximately 20% of patients with CUP, there is not any prognostic significance [26]. Generally, RAS-driven cancers are considered to be among the most difficult to treat; however, they are potentially targetable with MEK inhibitors [28]. To assess whether sensitivity to trametinib could be predicted in CUP cases, as well as to provide a tool to stratify patients for trials, an original “trametinib response signature” has been described in the literature [29]. This signature anticipated the experimentally assessed response to trametinib in agnospheres and was retrieved also in the matched patients’ tissues. It finally predicted the response in a retrospective cohort of CUP cases. Interestingly, CUP sensitivity predicted by the trametinib signature approximates that of BRAF-mutated melanoma. Although less frequent, *BRAF V600E* mutations were found in 1.6% (7 out of 442) in a large series of patients with CUP [30]. Circulating tumour DNA revealed that 80% of CUP patients (353 out of 442) had detectable alterations and 66% (290 out of 442) had at least one characterised alteration in the above-mentioned case series. Among these patients, alterations in MAPK and PI3K signalling were identified in 31.2% and 18.1%, respectively [30].

Genomic DNA is continuously confronted with a large number of DNA lesions. It is required for the cells to counteract DNA damage by activating the DNA damage response (DDR) in view of keeping the genome stable and securing cellular homeostasis [31]. Several DDR pathways have evolved in cells to repair different types of damage. *BRCA1* and *BRCA2* tumour suppressor genes play an important role in DDR, and mutations in these genes confer a high risk of breast and ovarian cancers [32,33,34,35]. *BRCA1* mutation carriers are at high risk of CUP (relative risk (RR) 3.45, 95% CI 2.35–5.07, *p* < 0.001) [36].

### 4.3. Angiogenesis

A cancer cell must be able to split from the main tumour; penetrate through surrounding tissues and basement membranes; and then enter and survive in the circulation, lymphatics, or peritoneal space to colonise a secondary location. This is followed by extravasation into surrounding tissue, survival in the alien milieu, proliferation, and angiogenesis activation, all while avoiding apoptosis or an immune response [37]. Angiogenesis includes several stages, such as proteolytic degradation of the basement membrane and surrounding extracellular matrix, endothelial cell proliferation and migration, and finally tube formation [38]. Cancer cells rely heavily on this pathway for development, survival, and invasion. The activation of an angiogenic switch is essential for a lesion to expand above a certain length [21,39]. It seems that CUP presents an angiogenic incompetence at the primary site that limits the development of the primary tumour. Endogenous positive angiogenic factors include vascular endothelial growth factor (VEGF), platelet-derived growth factor (PDGF), fibroblast growth factors (FGFs), epidermal growth factor (EGF), transforming growth factor (TGF), matrix metalloproteinases (MMPs), tumour necrosis factor (TNF), and angiopoietins, whereas endogenous negative angiogenic factors are interleukins, interferon, tissue inhibitors of metalloproteinases (TIMP), angiostatin, and endostatins [40]. The role of the angiogenesis within the biology of CUP is supported by the observation of its absence in primary tumours inducing dormancy, whilst it is present at metastatic sites. However, VEGF expression is not associated with prognosis, excluding the positive association between VEGF and the density of micro-vessels. A study reported that regardless of the overexpression of VEGF in 26% of a CUP case series, there was not any prognostic impact of CD34 and VEGF on the survival [41].

Similarly, the comparison between 39 liver metastases from patients with CUP versus 30 liver metastases from colon and breast cancer did not reveal differences in the density of micro-vessels; both groups exhibited high angiogenic activity [42]. Finally, a study demonstrated low expression of VEGF protein in patients with CUP. Fifty patients with squamous carcinomas metastatic to the cervical lymph nodes were compared with 52 patients with metastases from a known primary. The authors proposed a pattern independent of angiogenesis of metastatic spread for squamous CUP metastasising to the cervical lymph nodes [43].

### 4.4. Evasion of Immune Destruction

Tumours avoid immune surveillance by generating immunosuppressive cytokines, including TGF-β. TGF-β has been shown to selectively block the production of five cytolytic gene products, namely, perforin, granzyme A, granzyme B, Fas ligand, and interferon-γ, which are together involved in cytotoxic T-lymphocytes (CTL)-mediated tumour cytotoxicity. TGF-β-activated Smad and ATF1 transcription factors bind to their promoter regions, repressing granzyme B and interferon-γ [44]. In a study, programmed cell death-1 (PD-1) expression was detected in the tumour-infiltrating lymphocytes of 58.7% of patients with CUP, whereas programmed death-ligand 1 (PD-L1) expression was found in 22.5% of the CUP specimens [45]. Within the context of the immune microenvironment markers, tumour mutation load and microsatellite instability were high in 11.8% and 1.8% of CUP patients, respectively [45]. Microsatellite instability was associated with a high tumour mutational burden and represented a predictive biomarker of response to immune checkpoint inhibitors in several malignancies [46]. Plasma-based circulating cell-free DNA (cfDNA) assays identified mutations in the DDR protein MLH1 (mutL homologue 1) in 1.6% of CUP patients [30].

## 5. Classification of CUP

The minority of patients with CUP (15–20%) present with clinical and pathological features that can be attributed to a primary culprit (Table 1). The favourable risk cancer subgroup comprises peritoneal adenocarcinomatosis of a serous papillary subtype, isolated axillary nodal metastases in females, squamous cell carcinoma involving nonsupraclavicular cervical lymph nodes, single metastatic deposit from unknown primary, neuroendocrine carcinomas of unknown primary, and men with blastic bone metastases and elevated prostate-specific antigen (PSA). The treatment of these patients is compatible with the corresponding primary guidelines for metastatic disease. Currently, new favourable subsets of CUP have emerged, including colorectal, lung, and renal CUP, which underly specific treatments [47]. These patients generally harbour a chemosensitive disease and, as such, longer life expectancy. 

The remaining 80–85% of CUP patients are assigned to the unfavourable subset comprise two prognostic groups, according to the performance status (0 or 1) and lactate dehydrogenase (LDH) level [48]. These patients do not respond well to the empiric broad-spectrum chemotherapy and therefore the median overall survival is approximately 6–10 months. As far as the unfavourable subset is concerned, the one-year survival rates in good- and poor-risk patients are 53% and 23%, respectively.

## 6. Diagnostic Workup

The diagnosis of CUP is established when a metastatic cancer is histologically confirmed in the absence of identifiable primary tumour site, despite the extensive diagnostic evaluation. Recent research has focused on using genomics and transcriptomics to identify the origin of the primary tumour, but it is still not always performed, especially in low-resource environments [49]. The development of tissue of origin classifiers for the analysis and diagnostics of CUP using a whole genome sequencing dataset of both primary and metastatic tumours is still an effort in progress [4].

### 6.1. Pathology and Immunohistochemistry

From the histological perspective, CUP is defined as well- or moderately differentiated adenocarcinomas, accounting for 50–70% of all cases, with poorly differentiated carcinomas and adenocarcinomas making up another 20–30%, and the remaining being squamous-cell carcinomas (5–8%) and undifferentiated malignant neoplasms (2–3%) (Figure 1) with inability of light microscopy to distinguish among carcinomas, lymphomas, melanomas, and sarcomas [47,50,51,52]. The diagnoses of neuroendocrine tumours, melanomas, and sarcomas can be based on immunoperoxidase staining.

Indeed, standard morphology and IHC remain the main strategy for the identification of the primary tumour in patients with CUP. The technique involves the analysis of tissue sections with antibodies against particular tumour-specific antigens, structural tissue components, hormonal receptors, hormones, or antigens (Table 2). In the first instance, IHC differentiates well- and moderately-differentiated adenocarcinomas, squamous cell carcinomas, carcinomas with neuroendocrine differentiation, poorly differentiated carcinomas, and undifferentiated neoplasms. In squamous cell carcinomas and neuroendocrine carcinomas, the use of cell differentiation markers is advised, especially when tumour morphology is heterogenous or poorly differentiated. The IHC detection of markers, such as vimentin, S100 family proteins, HMB45 antigen, or CD45, may classify part of CUPs as non-carcinomas—sarcomas, melanomas, or lymphomas—that can be treated appropriately [53]. Nevertheless, in the CUPISCO trial, the misdiagnosis of CUP due to sarcomas and melanomas represented 1.6% and 5.6%, respectively, of the failure cases [54]. The most commonly used markers for the staining of CUP are the keratin family members, CK7 and CK20, with CK7+/CK20− being the most common in CUP (Figure 2). In CK7+/CK20− cases, ER positivity, and GATA3 positivity in ER-negative cases primarily direct clinicians to the breast as a possible site of cancer origin, especially in patients with axillary lymph node metastases [4]. Although TTF1 expression in a metastatic setting does not unquestionably prove a primary origin in the lung, all TTF1/napsin A-positive cases should be radiologically investigated to rule out lung primitivity. The expression of PAX8/WT1 is considered in order to investigate a possible gynaecological origin. Overall, in approximately one-third of CUP cases, the primary site is identifiable through the IHC staining panels [55]. However, a consensus panel of IHC markers has not yet been established, whilst no single pathognomonic marker exists for a conclusive diagnosis. Moreover, IHC has limited value in the diagnosis of poorly differentiated cancers. When immunoperoxidase stains are inconclusive in young patients with poorly differentiated tumours, electron microscopy should be considered in their evaluation. The presence of pleomorphic neoplastic cells with cytoplasmic vacuolations and/or cytoplasmic, non-membrane bound, electron-dense deposits detected by electron microscopy may suggest the diagnosis of a poorly differentiated carcinomas. Cytogenetic studies may be useful for the evaluation of young patients with poorly differentiated carcinomas or undifferentiated lesions that are responsive to chemotherapy. Finally, neurosecretory granules are specifically detected by electron microscopy in neuroendocrine tumours. Limitations of the IHC are the lack of reproducibility, due to the IHC-generated preparations, along with methodological issues. Failure to ensure that samples are of a high quality can hinder subsequent image analysis processes, negatively impact on data quality, and in some cases prevent an image analysis study from proceeding. Research using digitised histopathology slides for the development of artificial intelligence algorithms has increased markedly over recent years.

### 6.2. Diagnostic Radiology

Image-assisted technologies has revolutionised the diagnosis of CUP. CT and conventional MRI have both been used to locate lesions, considering the clinical manifestation of CUP. The diagnostic accuracy of CT scans is around 55% (36–74%), mainly in pancreatic, colorectal, and lung cancer, while MRI has a sensitivity of 70% in detecting primary breast cancers [56]. However, the diagnosis can be challenging if the primary tumour is small in size or has regressed, hindering the diagnosis. In particular, these cases may be successfully facilitated with the 2-[^18^F] fluoro-2-deoxy-d-glucose (FDG) PET/CT, but still the detection rate is around 40% [57]. The most frequent primary sites identified by PET are lung (33%) and head and neck (27%), followed by pancreas, breast, and colon (4–5%). Finally, ^68^Ga-DOTA-NOC receptor PET/CT is recommended for the identification of primary neuroendocrine tumours, along with their metastases [58]. Mammography is recommended for female patients with metastatic adenocarcinomas involving axillary lymph nodes. In patients with mammographically occult breast cancer, breast MRI may be considered.

### 6.3. Endoscopy

Endoscopy should be directed towards investigating specific symptoms and signs or when specific histopathological findings are available. Fiberoptic bronchoscopy is reasonable for patients with respiratory symptoms and/or expression of CK7 and TTF1, whereas colonoscopy should be requested for those with abdominal symptoms or occult blood in the stool and/or expression of CK7, CK20, and CDX2. The sensitivity and specificity of the endoscopies are generally low.

### 6.4. Serum Tumour Markers

In almost 70% of CUP patients, more than one marker can be concomitantly elevated in a non-specific way. Routine request of cancer antigen 125 (CA 125), cancer antigen 15.3 (CA 15-3), carbohydrate antigen 19-9 (CA 19-9), and carcinoembryonic antigen (CEA) is not recommended due to lack of prognostic and/or predictive value [59]. However, there are some clinical scenarios in which serum tumour markers may have some diagnostic value. Indeed, serum PSA should be evaluated in men with osteoblastic bone metastases, CA 125 in women with primary serous papillary peritoneal adenocarcinoma, and CA 15-3 in females with isolated axillary adenocarcinoma. Finally, a high level of thyroglobulin in patients with CUP and bone metastasis may indicate occult thyroid cancer [60].

### 6.5. Liquid Biopsy

Improvements in nucleic acid sequencing technologies have enabled the detection of low quantities of tumour genetic material within the blood and show the potential to be both sensitive and specific to an individual’s tumour. These blood-based biomarkers include cfDNA, tumour microRNAs (miRNAs), and platelet-derived tumour mRNA, as well as analysis of DNA, RNAs, and protein expression from individual circulating tumour cells. Within this context, the use of liquid biopsies reduces the need for intrusive diagnostic biopsies and provides enough material to perform the diagnostic procedures. For instance, even though the presence of aberrant hypermethylation of tumour-suppressor genes in serum DNA has been detectable before the millennium, more sensitive and quantitative techniques for analysis of DNA methylation are required to expedite its incorporation in the clinical setting [61]. In diffuse large-B-cell lymphoma, detection of aberrant DAPK1 methylation in cfDNA at the time of diagnosis is a positive prognostic biomarker, whilst in hepatocellular carcinoma, methylation of *VIM* is an early detection biomarker [62,63]. Overall, there is evidence that the tissue of origin can be determined using cfDNA [64]. 

### 6.6. Molecular Profiling for the Tissue of Origin

Molecular profiling technologies including microarray-based gene expression profiling, reverse transcriptase polymerase chain reaction, RNA sequencing, somatic gene mutation profiling with NGS, and DNA methylation profiling were used to define the primary culprit among patients with CUP. However, the implementation of tissue-of-origin classifiers in CUP is limited due to the absence of primary tumour. Some studies were conducted to validate predictions of the primary origin, on the basis of autopsy data, latent primary emergence, or IHC. Gene expression profiling has been directly compared to IHC, within known metastatic tumour types. Accuracy of gene expression profiling was 89%, compared with 83% for IHC when only one round of IHC determined the diagnosis, whereas in poorly differentiated cancers, such as CUP, the percentages were 83% and 67%, respectively [65]. Nevertheless, only a limited number of studies have investigated the clinical outcomes of CUP patients, treated on the basis of gene expression predictions [66]. Predicting the tissue-of-origin via molecular profiling is a debated topic within CUP, given that it is difficult to be molecularly classified in the absence of histological definition. However, the incorporation of molecular classifiers to the standard diagnostic workup may potentially identify atypical presentations of patients for whom site-specific therapies would be effective [67]. 

PlexinB2 (PlxnB2) is a semaphorin receptor implicated in the regulation of cancer cell proliferation, invasiveness, and metastatic spreading [68,69,70,71]. The G842C-PlxnB2 variant has been investigated in an effort to establish a proof of principle about the relevance of axon guidance genes in CUP. This mutation affected the conserved fold of an IPT domain, a moiety also found in Met and Ron oncogenic receptors [72,73]. Notably, the large intracellular portion of the plexins does not contain a kinase domain or other classical signalling domains; nevertheless, it regulates the activity of monomeric GTPases, especially R-Ras, Rap-1, and RhoA. Moreover, plexins have been shown to couple with transmembrane tyrosine kinases such as ErbB2 and Met, triggering alternative noncanonical signalling cascades, especially in cancer cells [74]. A recent study demonstrated that G842C-mutated PlxnB2 was competent for signalling, even in the absence of semaphorin stimulation [75]. Moreover, although knocking down PlxnB2 expression in CUP cells bearing a wild-type receptor had no any functional impact, the metastatic cells carrying the G842C mutation were found to be dependent on this variant PlxnB2 to sustain self-renewal and proliferation in culture, along with tumorigenesis in mice. These data indicated that G842C-PlxnB2 may be considered a gain-of-function mutation. Members of the tyrosine kinase receptor family have been associated with plexin signalling in cancer cells. According to the study, PlxnB2 was found in complex with EGFR, and EGFR phosphorylation was enhanced in the presence of G842C-PlxnB2. Moreover, the greater invasiveness of CUP cells driven by the expression of the mutated plexin was abrogated by selective EGFR inhibitors, namely, cetuximab and erlotinib. These data provide evidence of the functional involvement of an unexpected aberrant signalling pathway in CUP development and prompt for the characterisation of additional axon guidance mutated genes in CUP. 

## 7. Treatment of CUP

Traditionally, CUP patients who are classified into one of the favourable subsets are treated according to their corresponding primary guidelines for metastatic disease. CUP patients with poorly differentiated carcinoma with midline distribution (extragonadal germ cell syndrome) should be managed like poor prognosis germ cell tumours with platinum-based combination chemotherapy. More than 50% response has been reported, with 15–25% complete responders and 10–15% long-term disease-free survivors. Women with papillary adenocarcinoma of the peritoneal cavity are optimally treated as FIGO stage III ovarian cancer. The recommended strategy includes aggressive surgical cytoreduction, followed by platinum-based postoperative chemotherapy. The median response rate is 80%, whilst 30–40% of the patients are complete responders. Similarly to FIGO stage III ovarian cancer patients, the median survival is 36 months [76,77]. For the subgroup of women with adenocarcinoma involving only axillary lymph nodes, locoregional treatment with or without systemic therapy is suggested. The management is compatible with stage II/III breast cancer, resulting in 5- and 10-year overall survival rates of 75 and 60%, respectively. The patients with squamous cell carcinoma involving cervical lymph nodes are treated with locoregional management, according to the guidelines for locally advanced head and neck cancer. The 5-year survival rates range from 35 to 50% with documented long-term disease-free survivors. Surgery alone is inferior and only recommended in selected patients, particularly those with pN1 neck disease with no extracapsular extension. Radiotherapy to the ipsilateral cervical nodes alone is still inferior to extensive irradiation to both sides of the neck and the mucosa in the entire pharyngeal axis and larynx. Whether such intensive irradiation prolongs survival is still uncertain. Although the role of systemic chemotherapy remains undefined, concurrent chemoradiotherapy seems to be beneficial, particularly in patients with an N2 or N3 lymph node disease. The group of CUP patients with poorly differentiated neuroendocrine carcinomas should be treated with empirical platinum-based or platinum-taxane chemotherapy. The reported response rates are as high as 50–70% with 25% complete responders and 10–15% long-term survivors. Men with blastic bone metastases and elevated PSA are considered as having advanced/metastatic prostate cancer and treated accordingly. The appropriate approach of CUP patients with a single small metastasis is the local treatment with either resection and/or radiotherapy. A considerable number of these patients have a long disease-free survival [78]. Finally, the treatment of Merkel cell cancer (MCC) of unknown origin is largely multimodal in nature and includes surgery, radiotherapy, and chemotherapy. For primary MCC that is associated with clinically positive nodal disease or with positive sentinel node, complete dissection of the involved regional nodal basin is recommended [79]. MCC is radiosensitive, and as such, radiotherapy may be an alternative definitive treatment for medically ineligible surgical resection patients. In contrast, adjuvant chemotherapy has a limited role in MCC.

The treatment of patients with unfavourable CUP subsets is usually empirical chemotherapy, consisting of either taxanes or platinum-based regimens, on the basis of randomised trials showing dismal survival improvements [80]. The biomarker-based approach has been considered using targeted-therapy; nevertheless, the available evidence is limited to anecdotal cases [81]. Site-specific therapy guided by molecular classifiers was evaluated in this context. A meta-analysis of two retrospective and two prospective trials evaluating site-specific treatments in CUP was performed [82]. A trend towards improved overall survival was noted with site-specific versus empiric treatment for CUP (hazard ratio (HR) = 0.73, 95% confidence interval (CI) 0.52–1.02). The results of this meta-analysis highlighted the significant heterogeneity between the prospective studies comparing molecularly tailored to empiric therapy for CUP. In the most up-to-date meta-analysis of five studies that included 1114 patients, site-specific therapy was not significantly associated with improved overall survival (HR 0.75, 95% CI 0.55–1.03, *p* = 0.069) compared with empiric therapy [83].

CUPISCO (NCT03498521) is an ongoing prospective, phase II, randomised study designed to elucidate the potential benefit of treatment following genomic profiling, as compared to standard chemotherapy of CUP patients [84]. The study includes an atezolizumab monotherapy arm for the tumour mutational burden-high patients and a combination chemotherapy/atezolizumab arm for patients with tumour mutational burden-low or unknown tumours. The study experienced severe issues in patients’ accrual, along with screen failures. Molecular analyses, such as the identification of currently non-targetable alterations that may affect disease dynamics or be correlated with resistance, should be performed. Since CUP is clinically and molecularly heterogeneous, it would be reasonable to establish master protocols for enhancing the clinical trial strategy and direct patients to individually tailored treatment. I-PREDICT is an ongoing study that recruits patients with treatment-refractory solid tumours, including CUP, managed with individualised treatment, on the basis of genomic profiling (NCT02534675) [85]. Patients treated with matched therapy that impacted more than half of their genomic alterations achieved significantly better outcomes than those from the lower match group.

Within this context, immunotherapies have the potential to improve outcomes in this population, due to the PD-L1 expression and high tumour mutational burden in 22.5% and 11.8% of CUP patients, respectively [45]. Overall, the genomic mutation correlates of response and resistance to immune checkpoint inhibitors do not differ between CUP and tumours that are immune checkpoint inhibitor eligible [86]. Tumour mutational burden >10 mutations per megabase trended towards better outcomes in CUP patients treated with immune checkpoint inhibitors. Furthermore, MDM2 amplification, which is associated with lack of response to immune checkpoint inhibitors, has been detected in 2% of CUP patients [45].

Initially, some anecdotal cases showed clinical activity of immune checkpoint inhibitors in CUP, irrespective of the presumed tissue of origin [30]. Throughout time, we understood that the immune profiling of CUP is similar to that of malignancies responsive to immune checkpoint inhibitors and as such several trials investigate their efficacy in CUP [87]. The phase II NCT03391973 and NCT03752333 trials of pembrolizumab are currently in progress, whereas NivoCUP, an open-label phase II study, has already demonstrated a clinical benefit of nivolumab in CUP patients [88]. The reported objective response rate (ORR) was 22.2% in 45 previously treated patients, which met the primary endpoint. In 5 out of 12 patients who achieved a partial or complete response, the duration of the response was longer than 6 months. In the same subset of previously treated patients, the median overall survival was 15.9 months, whilst in the entire population of chemotherapy-naïve and previously treated patients, the ORR and the median overall survival were 21.4% and 16.2 months, respectively. These data provide evidence that nivolumab should be further investigated and may be incorporated in the therapeutic strategy of CUP. In the same study, a very low number of patients were treated upfront with nivolumab with 18.2% ORR. Overall, there is strong evidence that identification of predictive biomarkers is crucial in order to identify this one-fifth of CUP patients who may potentially respond to immune checkpoint inhibitors.

## 8. Conclusions and Future Directions

CUP is a heterogeneous group of metastatic tumours with a distinct natural history that mainly depends on clinicopathological criteria. While favourable groups are treated according to their corresponding primary tumour, unfavourable groups are treated with empirical chemotherapy, usually having a dismal prognosis. Several tissue-of-origin classifiers have been developed, collecting evidence that supported their translational potential in the clinical management of CUP patients. Several studies focused on genomic analysis of ctDNA and included some CUP cases among other tumour types, showing high sensitivity rates in the identification of oncogenic and actionable alterations in CUP. Small non-coding RNAs and epigenetic modifications are particularly appealing. Such biomarkers could potentially endorse the access to more specific therapies. The knowledge of the primary site remains fundamental because specific driver mutations could be predictive of responses in some tumour types but not in others. Immunotherapy is emerging as a potentially winning therapeutic strategy in several cancer types. Reasonably, it has gained interest even in the subset of CUP patients. Liquid biopsy could help in unveiling druggable alterations using a non-invasive approach. Therefore, molecular diagnostics, combined with genetic profiling, might become the standard of care for future CUP management. There is a significant limitation of the research on therapeutic strategies in CUP; this is the non-inclusion of many patients within the expanded favourable CUP subsets in the randomised trials who may be treated by their oncologists according to a potential primary tumour. New comprehensive clinical trial designs have been proposed to overcome the methodological issues encountered in CUP research implementing the latest diagnostics and therapeutic advances of CUP research.

## Figures and Tables

**Figure 1 ijms-24-05588-f001:**
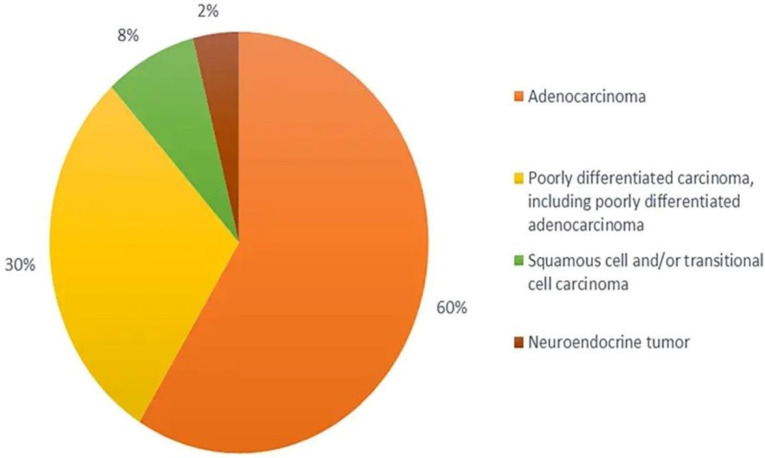
Histological categorisation of CUP.

**Figure 2 ijms-24-05588-f002:**
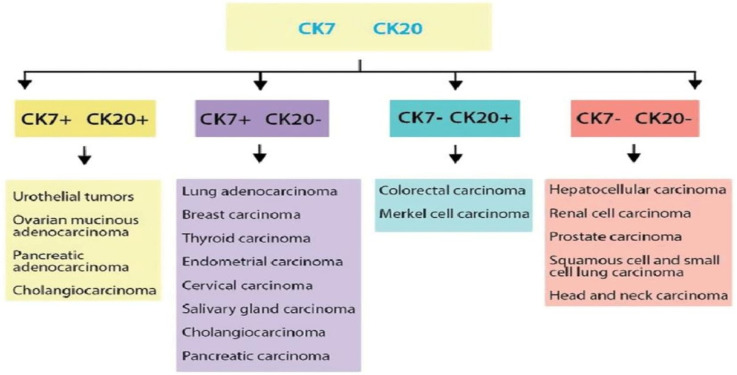
Cytokeratins used in CUP.

**Table 1 ijms-24-05588-t001:** List of the favourable and unfavourable subtypes.

	Favourable Subsets	Unfavourable Subsets
1	Poorly differentiated carcinoma with midline distribution (extragonadal germ cell syndrome)	Adenocarcinoma metastatic to the liver or other organs
2	Women with papillary adenocarcinoma of the peritoneal cavity	Non-papillary malignant ascites (adenocarcinoma)
3	Women with adenocarcinoma involving only axillary lymph nodes	Multiple cerebral metastases (adeno or squamous carcinoma)
4	Squamous cell carcinoma involving cervical lymph nodes	Multiple lung/pleural metastases (adenocarcinoma)
5	Isolated inguinal adenopathy (squamous carcinoma)	Multiple metastatic bone disease (adenocarcinoma)
6	Poorly differentiated neuroendocrine carcinomas	Squamous abdominopelvic CUP
7	Men with blastic bone metastases and elevated PSA (adenocarcinoma)	
8	Patients with a single, small, potentially resectable tumour	
9	CUP patients with a single small metastasis	
10	Merkel cell adenopathy of unknown origin	

Abbreviations: CUP; cancer of unknown primary, PSA; prostate-specific antigen.

**Table 2 ijms-24-05588-t002:** Immunohistochemistry tests for investigating CUP.

Tumour Type	Immunoperoxidase Marker
Carcinoma	Cytokeratins, EMA
Lymphoma	CLA, EMA
Sarcoma	Vimentin, desmin, factor VIII antigen
Melanoma	S-100, HMB-45, NSE, vimentin
Neuroendocrine	Chromogranin, EMA, NSE, cytokeratins, synaptophysin
Germ cell	hCG, AFP, EMA, cytokeratins
Prostate cancer	PSA, EMA, cytokeratins (CK7−/CK20−)
Breast cancer	ER, PR, EMA, cytokeratins (CK7+/CK20−)
Thyroid cancer	Thyroglobulin, cytokeratins (CK7+/CK20−), calcitonin, EMA

Abbreviations: CUP; cancer of unknown primary, EMA; epithelial membrane antigen, CLA; cutaneous lymphocyte antigen, NSE; neuron-specific enolase, hCG; human chorionic gonadotrophin, AFP; alpha fetoprotein, PSA; prostate-specific antigen, ER; estrogen receptor; PR; progesterone receptor.

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
