# Peer review of "From Biology to Diagnosis and Treatment: The Ariadne’s Thread in Cancer of Unknown Primary"

_ijms, 2023, doi:10.3390/ijms24065588_

Round 1

Reviewer 1 Report

The authors dedicate this paper to the cancer of unknown primary (CUP) group. The work is of extreme interest since, not knowing the initial causes, it is difficult to make a prognosis and, in a certain way, a good diagnosis.

The state of the art and literature review is very well done. However, it would be interesting to add more data, or make them more visible in the work. That is, in terms of percentage, how many people are we talking about? what type of cancer?

Still, for such an extensive and detailed review, a more robust conclusion is missing. What future work do you suggest?

If some figure could be added, the work would not become so heavy in reading.

Author Response

Dear Editor and Reviewers,

I am pleased to resubmit for publication the revised version of ijms-2237706 manuscript, entitled “From Biology to Diagnosis and Treatment: The Ariadne's Thread in Cancer of Unknown Primary”.

Thankfully the reviewers provided us with a great deal of guidance, regarding how to better position the article. We are hopeful you agree that this revision will update our comprehensive review. All the comments have been addressed, as shown in the revised version of the manuscript, along with this point-by-point response to the reviewers' comments.

All corresponding are blue changes in the manuscript.

Reviewer #1:

The authors dedicate this paper to the cancer of unknown primary (CUP) group. The work is of extreme interest since, not knowing the initial causes, it is difficult to make a prognosis and, in a certain way, a good diagnosis.

The state of the art and literature review is very well done. However, it would be interesting to add more data, or make them more visible in the work. That is, in terms of percentage, how many people are we talking about? what type of cancer?”.

Response:

Thank you very much for your kind words about our paper. We appreciate the opportunity to revise our work for consideration for publication.

In terms of your comment about the “percentage, how many people are we talking about? what type of cancer?”, we refer you to the introduction (lines 55-57), epidemiology (lines 78-94) and risk factors (lines 96-106) sections, respectively. We believe that we adequately cover your concerns in these parts of our manuscript.

Still, for such an extensive and detailed review, a more robust conclusion is missing. What future work do you suggest?”

Response:

Thank you for your consideration. We have updated the conclusion section, which currently includes “future directions” (lines 496-517 of the revised manuscript).

If some figure could be added, the work would not become so heavy in reading.”

Response:

Thank you for your recommendation. We have now added figures 1 and 2 (lines 267-268 and 309-310, respectively).

Reviewer 2 Report

I am very impressed with the manuscript. The manuscript is well organized and systematic written. The information is quite concrete. However, I suggest to write and elaborate more about the future directions and clinical significance of the study. I would also suggest authors to provide some systematic diagrams to attract readers. Otherwise, I believe the current format of the manuscript is suitable for publication.

Author Response

Dear Editor and Reviewers,

I am pleased to resubmit for publication the revised version of ijms-2237706 manuscript, entitled “From Biology to Diagnosis and Treatment: The Ariadne's Thread in Cancer of Unknown Primary”.

Thankfully the reviewers provided us with a great deal of guidance, regarding how to better position the article. We are hopeful you agree that this revision will update our comprehensive review. All the comments have been addressed, as shown in the revised version of the manuscript, along with this point-by-point response to the reviewers' comments.

All corresponding are blue changes in the manuscript.

Reviewer #2:

I am very impressed with the manuscript. The manuscript is well organized and systematic written. The information is quite concrete. However, I suggest to write and elaborate more about the future directions and clinical significance of the study.”.

Response:

Thank you for your positive reinforcement. We appreciate the opportunity to revise our work for consideration for publication.

We have updated the conclusions to “conclusion and future directions” section, highlighting the key messages of the study.

I would also suggest authors to provide some systematic diagrams to attract readers. Otherwise, I believe the current format of the manuscript is suitable for publication.

Response:

Once again, we express our gratitude for your positive feedback. Otherwise, we have incorporated figures 1 and 2 for a more attractive for the readers vision of the manuscript.

Reviewer 3 Report

The manuscript by Beatrice Gadiel Mathew and collaborators aims to review the recent literature on CUPs to describe the biology, the molecular profile, the classification of this class of carcinomas.

CUPs represent a class of enigmatic metastatic carcinomas for which it is impossible to identify a primary site of origin or it is impossible to establish a primary tissue of origin. The theme of work is current and in line with the aims of the magazine.

I have Major Revisions

Line 164: Usually melanomas or melanoma metastases are not included in the CUP since they are not "epithelial Carcinomas". Moreover, to date it is impossible to distinguish a cutaneous melanoma from a mucosal melanoma with the methods available to us. Melanoma metastases are more similar to occult melanomas than to CUP. It is advisable to delete the sentence that is not in line with the topic. Furthermore, there are manuscripts in the literature demonstrating the use of anti MEKs against CUPs. It is advisable to consider these manuscripts and describe this latest research (https://doi.org/10.1038/s41467-021-22643-w)

Line 259: For the same criteria specified above, lymphomas and sarcomas as well as melanomas cannot be considered CUP. Please rephrase the sentence.

Line 185: The chapter on angiogenesis has little focus on CUPs and should be revised. I suggest including the mechanism by which these tumors arise. There are several theories in the literature that explain how a totipotent cell can travel in the blood and lymphatic stream arriving in a tissue different from that of origin where it re-differentiates into a carcinoma of unknown origin.

Lines 264-276: I'm sorry but the description is confusing. IHC allows to exclude lymphomas, sarcomas and melanomas and, in cases of CUP, it allows to identify a tumor phenotype that recalls a tissue of origin (e.g. renal, pancreatic, pulmonary). Please rephrase. I suggest a manuscript to rearrange the concept (DOI: 10.1007/s00428-022-03435-z)

Line 281: The authors are asked to explain the role of electron microscopy in cases of poorly undifferentiated carcinomas.

Line 285: IHC is usually a standardized method. Furthermore, its interpretation must be made by specialists in the sector who should normally guarantee high reliability in the analysis of the slides. It is advisable to rephrase the sentence.

To implement the quality of the review there are more recent works that identify new mechanisms of progression in the CUPs. An example is the role of PLXNB2 in cancers of unknown primary. The authors may consider re-evaluating the more recent literature in this regard.

Minor revisions

The text must be formatted according to the criteria of the journal.

Line 68 and line 251:  please cite a recent manuscript identifying a simplified algorithm in CUP diagnostics in order to save valuable material to be used for molecular analyzes (DOI: 10.1007/s00428-022-03435-z)

Line 238 and line 386: please enter a references.

Author Response

Dear Editor and Reviewers,

I am pleased to resubmit for publication the revised version of ijms-2237706 manuscript, entitled “From Biology to Diagnosis and Treatment: The Ariadne's Thread in Cancer of Unknown Primary”.

Thankfully the reviewers provided us with a great deal of guidance, regarding how to better position the article. We are hopeful you agree that this revision will update our comprehensive review. All the comments have been addressed, as shown in the revised version of the manuscript, along with this point-by-point response to the reviewers' comments.

All corresponding are blue changes in the manuscript.

Reviewer #3:

  • General comments:

The manuscript by Beatrice Gadiel Mathew and collaborators aims to review the recent literature on CUPs to describe the biology, the molecular profile, the classification of this class of carcinomas.

CUPs represent a class of enigmatic metastatic carcinomas for which it is impossible to identify a primary site of origin or it is impossible to establish a primary tissue of origin. The theme of work is current and in line with the aims of the magazine.

I have Major Revisions”.

Response:

We appreciate you taking the time to offer us your comments and insights related to the paper. Thank you for your constructive feedback. We tried to be responsive to your concerns as we approached our revision.

  • Major revisions:

  1. Line 164: Usually melanomas or melanoma metastases are not included in the CUP since they are not "epithelial Carcinomas". Moreover, to date it is impossible to distinguish a cutaneous melanoma from a mucosal melanoma with the methods available to us. Melanoma metastases are more similar to occult melanomas than to CUP. It is advisable to delete the sentence that is not in line with the topic. Furthermore, there are manuscripts in the literature demonstrating the use of anti MEKs against CUPs. It is advisable to consider these manuscripts and describe this latest research (https://doi.org/10.1038/s41467-021-22643-w).

Response:

Thank you for your comment. We fully agree and we have deleted that sentence as you kindly recommended. We have also mentioned the “trametinib response signature” that very interestingly was described by Verginelli et al (lines 162-168 of the revised manuscript, reference 29).

  1. Line 259: For the same criteria specified above, lymphomas and sarcomas as well as melanomas cannot be considered CUP. Please rephrase the sentence.

Response:

Indeed, we agree and rephrased, appropriately (lines 259-265 of the revised manuscript).

  1. Line 185: The chapter on angiogenesis has little focus on CUPs and should be revised. I suggest including the mechanism by which these tumors arise. There are several theories in the literature that explain how a totipotent cell can travel in the blood and lymphatic stream arriving in a tissue different from that of origin where it re-differentiates into a carcinoma of unknown origin.

Response:

Thank you for your consideration. Although metastatic sites of CUP show a high degree of vascularisation, the available data do not show a specific biological role for angiogenesis in the metastatic phenotype of CUP.

We have reshaped that section by incorporating the data of the angiogenesis that were originally included in the “Biology of CUP” section We have also mentioned that it is hypothesised that CUP presents an angiogenic incompetence at the primary site, which thereby limits the development of the primary tumour (lines 183-193 of the revised manuscript).

  1. Lines 264-276: I'm sorry but the description is confusing. IHC allows to exclude lymphomas, sarcomas and melanomas and, in cases of CUP, it allows to identify a tumor phenotype that recalls a tissue of origin (e.g. renal, pancreatic, pulmonary). Please rephrase. I suggest a manuscript to rearrange the concept (DOI: 10.1007/s00428-022-03435-z).

Response:

Thank you for your recommendation and apologies for the confusion. We have rephrased and rearranged appropriately the context of this section. Moreover, we have added the references 4 and 54 (lines 273-290 of the revised manuscript).

  1. Line 281: The authors are asked to explain the role of electron microscopy in cases of poorly undifferentiated carcinomas.

Response:

Thank you for your suggestion. We have now explained the role of electron microscopy in cases of poorly undifferentiated carcinomas (lines 296-298 and 300-302 of the revised manuscript).

  1. Line 285: IHC is usually a standardized method. Furthermore, its interpretation must be made by specialists in the sector who should normally guarantee high reliability in the analysis of the slides. It is advisable to rephrase the sentence.

Response:

Thank you for your comment. We do agree that IHC interpretation should be made by specialists in the sector. At that point, we have rephrased accordingly, providing the challenges and we importantly mention the development of artificial intelligence algorithms (lines 303-307 of the revised manuscript).

  1. To implement the quality of the review there are more recent works that identify new mechanisms of progression in the CUPs. An example is the role of PLXNB2 in cancers of unknown primary. The authors may consider re-evaluating the more recent literature in this regard.

Response:

Thank you for your recommendation. We have now discussed the role of PLXNB2 in CUP in the section of molecular profiling for the tissue of origin and added the references 68-75 (lines 380-402 of the revised manuscript).

  • Minor revisions:

  1. The text must be formatted according to the criteria of the journal.

Response:

Thank you. We have taken into consideration journal’s format, but we will be advised further by the editorial office on that matter.

  1. Line 68 and line 251: please cite a recent manuscript identifying a simplified algorithm in CUP diagnostics in order to save valuable material to be used for molecular analyzes (DOI: 10.1007/s00428-022-03435-z).

Response:

Thank you for your comment; we have cited the reference 4 (lines 68 and 256 of the revised manuscript).

  1. Line 238 and line 386: please enter a references.

Response:

Thank you; we have entered references 48 and 78, respectively.

Round 2

Reviewer 3 Report

I thank the authors for making the requested changes which I believe make the manuscript much more complete.

I congratulate them for the excellent work done and I propose the publication of the manuscript in this new version.